# The Role of Non-Covalent Bonds in the Deformation Process of Coal: An Experimental Study on Bituminous Coal

**Hewu Liu** [1,*] **and Chenliang Hou** [2]

1   State Key Laboratory of Mining Response and Disaster Prevention and Control in Deep Coal Mines, Anhui University of Science and Technology, Huainan 232001, China
2   School of Resources and Earth Science, China University of Mining and Technology, Xuzhou 221116, China
*   Correspondence: liuhewu@aust.edu.cn

**Abstract:** The chemical structures of tectonically deformed coal are significantly altered by stress. However, the stress response of non-covalent bonds in deformation experiments and the role of non-covalent bonds in the deformation process of coal have not been studied yet. In this work, coals before and after simulative deformation experiments were systematically investigated to uncover the coal's deformation mechanism and the variation of non-covalent bonds. The results indicate that differential stress and temperature can promote ductile deformation while confine pressure hinders the deformation process. Differential stress and temperature in the ranges of 100–150 MPa and 100–200 °C, respectively, are key transition conditions from brittle to ductile deformation for the selected bituminous coal. Furthermore, hydrogen bonds and π–π bonds crosslinking coal molecular networks determine the mechanical properties of the coal. The simulative deformation experiments indicate that, with an increase in the coal's deformation intensity, hydrogen bonds and π–π bonds are inclined to be disrupted in the relaxation stage, which enhances the motion ability of the liberated molecular structures and reduces the brittleness of the coal. In the rearrangement stage, tighter and more ordered configurations are formed, accompanied by the formation of π–π bonds. Coals in the deformation experiments are inclined to undergo ductile deformation once sufficient non-covalent bonds are cleaved in the relaxation stage.

**Keywords:** non-covalent bonds; hydrogen bonds; π–π bonds; coal deformation; simulative deformation experiment



## 1. Introduction

The structural deformation of coals, which are both the source and reservoir of coalbed methane (CBM), has been extensively investigated from various perspectives for its significance in safe coal production and CBM exploitation [1–7]. Tectonically deformed coals (TDCs) with diverse deformation characteristics and mechanisms have been systematically classified into three typical categories, including brittle, brittle–ductile, and ductile deformation sequences [8,9]. Previous studies have shown that the development of brittle deformed coal is conducive to the exploitation of methane, while the development of the latter two enhanced the proneness of coal and gas outbursts [1,10]. With the significant progress of nanogeoscience [11,12], studies on the diverse types of TDCs have recently focused on brand-new nanoscale fields [8]. The stress response of coal's ultrastructure (nanoscale pores and macromolecular structures) has been further explored and clarified by geologists while researching natural and experimentally deformed coals [8,13–17]. These studies have shown that the occurrence of methane in a coal reservoir is intimately correlated with the evolution of the coal's macromolecular structures. Firstly, nanoscale pores surrounded by atoms are significantly modified by the stress-induced alteration of the inner chemical structures of coals, which in turn affects the permeability, pore size distribution and gas-bearing properties of the coal [8,16,18,19]. Secondly, the content of methane in

outburst coals (usually ductile deformed coals) is much higher than the detected values in normal, intact, bulk coal [2,20]. The origin of excess methane is also considered to be closely connected to the macromolecular evolution of TDCs from a mechanochemical aspect [2,21,22]. Therefore, investigations of coal deformation mechanisms and the evolution of macromolecules can shed light on the accumulation and occurrence of methane and enrich the theoretical basis of understanding the mechanisms of coal and gas outbursts.

The macromolecular evolution of coal is strongly unified with the macroscopic coal deformation induced by tectonic stress [2]. The early coalification process of TDCs advances with increasing deformation intensity (especially for ductile deformed coals), which has been summarized to mainly include stress degradation and polycondensation mechanisms [23,24]. More specifically, previous studies have shown that the stress degradation of macromolecular structures mainly includes aliphatic side chains and functional groups, while stress polycondensation mainly refers to the rearrangement and stacking of aromatic sheets [15,24–26]. Furthermore, under the influence of shear stress, the deformation of aromatic structures can directly lead to the fusion or generation of secondary structural defects [27–29].

So far, abundant simulative experiments have been performed to investigate the deformation mechanisms and evolution of the inner chemical structures of TDCs. In the 1990s, Ross and Bustin (1990), Bustin et al. (1995) and Ross and Bustin (1997) performed a series of creep experiments on anthracite with temperatures higher than 300 °C and pressures higher than 500 MPa to illustrate that non-hydrostatic stress (especially shear stress) could facilitate the graphitization process of anthracite [30–32]. Similarly, friction experiments with high velocity also demonstrated that the coal maturation degree is significantly advanced by the frictional heat generated by shear stress [33,34]. In coaxial sub-high temperature and pressure experiments, strain energy transformed from mechanical energy directly led to the disassociation of functional groups and the generation of secondary structural defects, especially in experiments with lower strain rates [21,35]. The stress could directly act on the inner macromolecular structures of coal, which ensures the cleavage and rotation of covalent chemical bonds [13,14].

Pioneering investigations on the evolution of coal macromolecular structures have focused on the alteration of covalent bonds, such as the elongation, rotation and cleavage of aromatic C-C, aliphatic C-C and C=O of oxygen functional groups [21,28,36]. However, the understanding of the stress response of coal macromolecules needs to be based on the knowledge of actual macromolecular structures. Firstly, as early as 1984, a two-phase model (non-associative model) was proposed by Shinn (1984) [37] to reveal the inner macromolecular structures of coal. The proposed model was mainly composed of covalently linked rigid macromolecular networks (immobile phase or host molecules) with relatively small molecules (mobile phase or guest molecules) trapped within [38–40]. While approaching the nature of coal molecular structures, researchers found that the non-associative model was not applicable to explain the extraction yields of coal in $CS_2$-NMP extraction experiments, which promoted the rise of associative and composite models and reminded researchers about the importance of non-covalent bonds [41–43].

In coal macromolecular networks, non-covalent bonds with a quadruple number of covalent cross-links play a key role in intersegment linking [44–46]. Non-covalent bonds in coal have been summarized to be mainly composed of ionic, hydrogen, and π-π bonds [47,48]. Among them, ionic bonds between ionized functional groups and cations are commonly present in lower-rank coal like lignite [49], which might be less significant in selected samples (bituminous coal) (Table 1). While the latter two are both considered to occur in bituminous coal [44,48,50], hydrogen bonds (HBs), as cross-links between small molecules and large molecules, become less significant with increasing coal rank for the disassociation of functional groups. Conversely, π–π interactions between aromatic structures increase with increasing coal rank. Additionally, the conformations of aromatic layers determine the π–π bonds, and a parallel configuration was in favor of the formation of π–π bonds. In the current study, the primary bituminous coal sample

was used for simulative experiments; this coal is considered to include both hydrogen and $\pi$–$\pi$ bonds [41,44,46,51]. Nevertheless, ionic bonds that are present in more numbers in lower-rank coals are ignored in this work [46,49].

**Table 1.** Ultimate and proximate analysis results of the raw coal sample.

| $R_{o\ max}$ | Proximate analysis (%) | | | | Ultimate Analysis (wt%) | | | | |
|---|---|---|---|---|---|---|---|---|---|
| | $M_{ad}$ | $A_d$ | $V_{daf}$ | $FC_d$ | $S_{t.d}$ | $O_{daf}$ | $C_{daf}$ | $H_{daf}$ | $N_{daf}$ |
| 0.9 | 0.9 | 4.5 | 37.6 | 59.6 | 0.3 | 8.3 | 84.4 | 5.5 | 1.5 |

Note: $R_{o\ max}$: maximum reflectance of vitrinite; $M_{ad}$: Inherent moisture content with air-dried basis; $A_d$: Ash yield with dry basis; $V_{daf}$: Volatile matter yield with dry-ash-free basis; $F_{Cd}$: Fixed carbon content with dry basis; ad: air-dried basis; d: dry basis; daf: dry-ash-free basis; $S_t$: total sulfur; O: Oxygen; C: carbon; H: hydrogen; N: nitrogen.

Coal properties, like rubbery transition, swollen behavior, pyridine extraction yield and molecular mobility, all depend on the existence of non-covalent bonds to some extent [45,50,51]. Among these properties, the mechanical properties, like the elasticity, brittleness and plasticity of coal, have all been demonstrated to be intimately correlated with the inner cross-linkages of the molecular network structures [52–54]. The results of these previous studies have shown that the breakage and formation of non-covalent bonds determine the mobility of small molecules, which in turn affects the mechanical properties of coal. Diverse types of treatments on coal can all lead to the disassociation of non-covalent bonds. The mobility of bituminous coal molecules or intersegments is enhanced by the disruption of HBs in the heating treatment [55]. Labile non-covalent bonds in coal can be easily disassociated under the influence of external factors. For example, previous studies have indicated that solvent treatment on coal relaxed the macromolecular network structures and increased the mobility of small molecules through the disruption of non-covalent bonds [56,57]. Similarly, at the microstructural level, the decrease of molecular chains entanglements and the increase in the mobility of molecules induced by heating treatment (<425 °C) directly lead to the increase in the plasticity of coal [58]. Furthermore, the stress-induced disruption of HBs is also an important mode that causes the relaxation of HB-dominated solid materials [59]. Our recent research results have shown that non-covalent bonds in natural coals can also be disassociated by tectonic stress [60]. Besides, drop weight impact testing has shown that non-covalent bonds are significantly disrupted by the transformed impact energy [61]. Moreover, shear stress can also directly breakup non-covalent bonds at the resolidification stage of coal carbonization, which changes the mechanical properties of the coal as well [58]. Proceeding from the bond strength, the disruption energies of hydrogen and $\pi$–$\pi$ bonds are one to two orders of magnitude lower than covalent bonds [41,60], which might be more sensitive to stress and heating treatment than covalent bonds. The deformation behavior of solids with cross-linked structures changes with the alteration of non-covalent bonds, for example, the wood, chitosan and nylon etc. [59,62,63]. However, the role of non-covalent bonds in coal deformation is unclear and needs further investigations.

Consequently, the variation of non-covalent bonds during the coal deformation process has rarely been studied by researchers, especially by using simulative experiments. In this study, to reveal the micro coal deformation mechanism, high temperature and pressure experiments were performed on bulk primary coal that was collected from Zhuxianzhuang colliery, Huaibei coalfield, China. The role of non-covalent bonds in the process of coal deformation was clarified by revealing the stress-induced alteration of non-covalent bonds in cross-linked macromolecular networks. By connecting the mechanical properties of coal to the evolution of non-covalent bonds, the coal deformation mechanism can be better revealed at the nano-scale.

## 2. Materials and Methods

### 2.1. Materials

2.1.1. Geological Setting

The Zhuxianzhuang colliery, which was selected as the sampling area, is located in the Suxian coal mine, which is an important coal production base in China, it lies at the southeast of Suzhou city, Anhui province (Figure 1a). Zhuxianzhuang colliery is in the East Suzhou syncline that belongs tectonically to the overlying systems of the Xisipo thrust fault. Reverse faults are the predominant fault structures in the Zhuxianzhuang colliery (Figure 1b), and they mainly control the distribution of TDCs in the local area [1]. The major minable coal seam 8, which was chosen for sampling, belongs to the lower Shihezi Formation of the Middle Permian, an important coal-bearing stratum in the Zhuxianzhuang colliery (Figure 1c). The Lower Shihezi Formation, which is a fluvial-dominated delta facies, is mainly composed of sandstone, mudstone and coal.

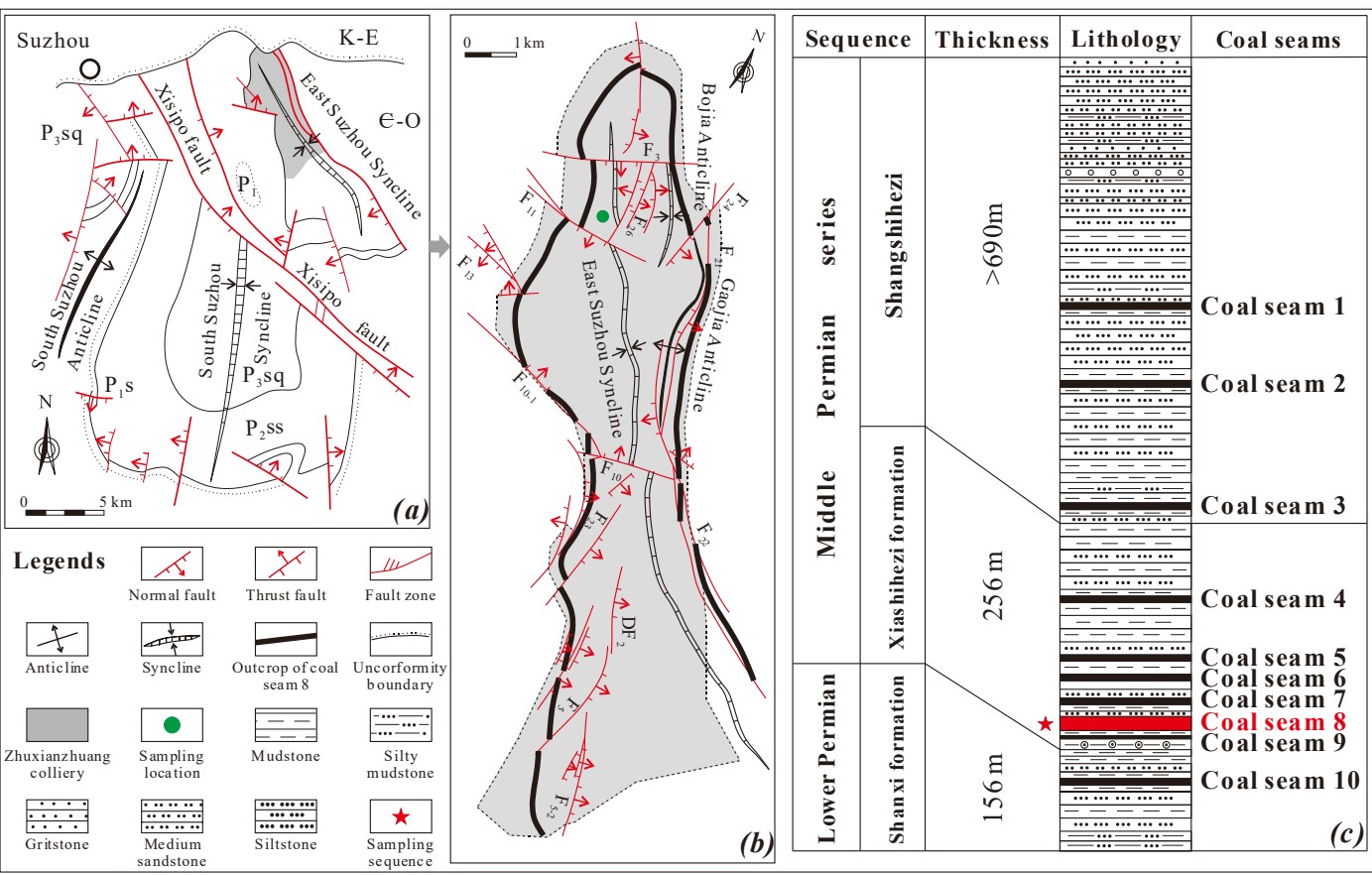

**Figure 1.** Distribution of sampling sites. (**a**) Structure outline of the Suxian coal mine and the location of the Zhuxianzhaung colliery. (**b**) Structure outline of the Zhuxianzhaung colliery and the location of the specific sampling site. (**c**) Detailed stratigraphic column of the coal-bearing strata in the Zhuanxianzhuang colliery (modified from Jiang et al. [1] and Liu et al. [60]).

2.1.2. Samples Preparation

The bulk coal sample collected had preserved primary structures in which maceral bands could be distinguished. Cleats and fractures could be barely identified in the bulk coal sample, which indicated that the collected bulk sample was less affected by tectonic stress since being deposited in the coal basin. According to the ASTM standard, the collected coal sample was classified as a high volatile bituminous coal after measuring the fixed carbon content and volatile matter yield (Table 1). The coal is characterized by low sulfur content (<2%), as illustrated by Chou et al. [64]. To simulate the tectonic stress

extrusion along the bedding plane, nine columnar samples with a diameter of 25 mm and a length of 40 mm were drilled parallel to the maceral bands utilizing the industrial drilling machine, Z4116.

### 2.1.3. Characterization of Coal Properties

The ultimate analyses of the coal samples were performed using a Vario Macro Cube element analyzer with an analysis precision of 0.05–0.15% and a detection range of 0.03% to 100%. The proximate analyses of coal samples strictly followed the GB/T 212-2008 standard.

The micro deformation behavior of coal samples before and after the deformation experiments was characterized using a High Solution 3D X-ray microanalyzer (XRM) with a tungsten target (Xradia 510 Versa, Carl Zeiss, Germany). The voltage of the X-ray tube was in the range of 30–160 kV; the X-ray power was 2–10 W, and the three-dimensional spatial resolution ranged from 55–0.5 μm. Firstly, the XRM data was processed using the trial version Dragonfly software (Montréal, QC, Canada) to acquire a reconstructed three-dimensional model. Then, the deformation characteristics before and after the deformation experiments were revealed by XRM slices along the *z*-axis of the three-dimensional reconstructed models and perpendicular to the maceral bands.

Based on the XRM images, the sites with typical deformation characteristic in the samples after the deformation were locked and selected as the sampling and testing objects. The characteristic of the molecular structures of the samples gathered from typical sites were detected by various test methods for the comparison of the molecular evolution characteristics under the influence of deformation and heating. The details and requirements of the detection methods are explained as follows.

FTIR (Fourier transform infrared spectroscopy) was used as a semi-quantitative method for assessing the HBs in coal [65]. The coal samples less than 75 μm sizes were firstly ground with KBr for 20 min, utilizing an agate mortar. Secondly, a mixture of KBr and coals was molded into a disc. Because the adsorption FTIR spectra of water molecules overlap with that of HBs in coal, the molded discs of the mixture should be vacuum-dried with a temperature of 105 °C to avoid the interference of water molecules in coal [66,67]. The selected Vertex 80 FT-IR spectrometer (manufactured by Bruker corporation, Karlsruhe, Germany) is equipped with a vacuum optical platform, which can completely protect the molded disks from water molecules. A mirror used as a background is considered to have not adsorbed water as well [65]. The beam diameter is usually around 40 mm, and the diaphragm is f/2.5. Pure ground KBr dried at a temperature of 150 °C was used as a reference.

XPS (X-ray photoelectron spectroscopy) is an important detection method to semi-quantitatively evaluate π–π bonds in coal [68,69]. Detections were performed on an Escalab 250Xi (manufactured by Thermo Fisher Scientific, Waltham, MA, USA) equipped with a source gun of monochromatic Al target Kα radiation. A 180° hemispheric energy analyzer was adopted during the test. The spot size of the X-ray was set to 900 μm; the energy step size was 0.05 eV, and the number of energy steps was 361.

Raman (Raman spectroscopy) was constantly applied to characterize the molecular structures of the coals [70]. The coal samples less than 75 μm sizes were detected utilizing Raman spectroscopy Senterra (manufactured by Bruker corporation, Karlsruhe, Germany) equipped with a three-dimensional automatic control platform. The exciting wavelength was 532 nm, and the laser power of the incident beam on the experimental samples' surfaces was maintained at 5 mW, and the spectral resolution was smaller than 1.5 cm$^{-1}$.

HRTEM (high-resolution transmission electron microscopy) is commonly applied to directly observe the aromatic structures of coal [71]. Ultrapure ethanol with dispersed powder coal samples was pipetted on lacy carbon films. After evaporating the ethanol, the remained organic nanoparticles of the coals were observed using a Tecnai G2 F20 (manufactured by FEI corporation, Hillsboro, OR, USA) equipped with STEM (200 kV) at the advanced analysis and computation center of the China university of mining and technology.

## 2.2. Methods

Simulative Experiment

The coal deformation experiments were performed using a TRTP-2000 coal high temperature and pressure deformation experimental system, which has been introduced in detail in Hou et al.'s research [72]. Five main parts constitute the experimental system, viz. deformation, pressure, cooling, heating, and controlling systems. The repeated calibration of the TRTP-2000 system revealed that 800 MPa and 800 °C were steadily attainable. Molded analytical pure NaCl was applied as the confining pressure medium. The migration speed of the axial pressure intender was controlled by a digital hydraulic servo. The axial and confine pressures in the experiments could be precisely controlled by the built-in actuators and pressure sensors. The programmed temperature was conducted using electric heating. A graphite tube was used as the heating medium. A thermocouple inserted in the axial pressure intender served to acquire the temperature of the heated samples. The regulator of the heating system was SHIMADEN SRS. The entire experimental system was kept open in the process of deformation, and gas diffusing from the experimental samples that affected the coal's mechanical properties in the system was not taken into account [32,73].

The drilled columnar sample was firstly inserted into the graphite tube, which was surrounded by consolidated NaCl (Figure 2). Then, the reaction kettle containing the coal sample was entirely placed in the pressure chamber. By controlling the motion of thr indenters in the pressure chamber, the samples and the consolidated NaCl were compressed to apply the axial and confine pressures for coal deformation, respectively. By applying a current to the indenters, the graphite tube was heated to achieve the heating of the samples. The heating and pressurizing processes were strictly made to adhere the following three stages to more realistically simulate the actual deformation of coal during geological processes. In the first stage, each columnar sample was synchronously loaded to the presetting temperature and the axial and confine pressures at rates of 1.5 °C/min and 0.5 MPa/min, respectively. To ensure the same heating time, the first stages of all the coals were maintained for 200 min. Additionally, the differential stress on the coals in the first stages was set to 0 MPa, that was, the axial pressure equaled the confine pressures in the first stage. Secondly, the coals with equivalent axial and confine pressures were continuously loaded with increasing axial stress (at a constant rate of 0.2 MPa/min) until the differential stress was satisfied with the preset value. Finally, the terminal differential stress and temperature were maintained for specified times during the creep experimental stage. The specific experimental conditions are shown in Table 2 in detail.

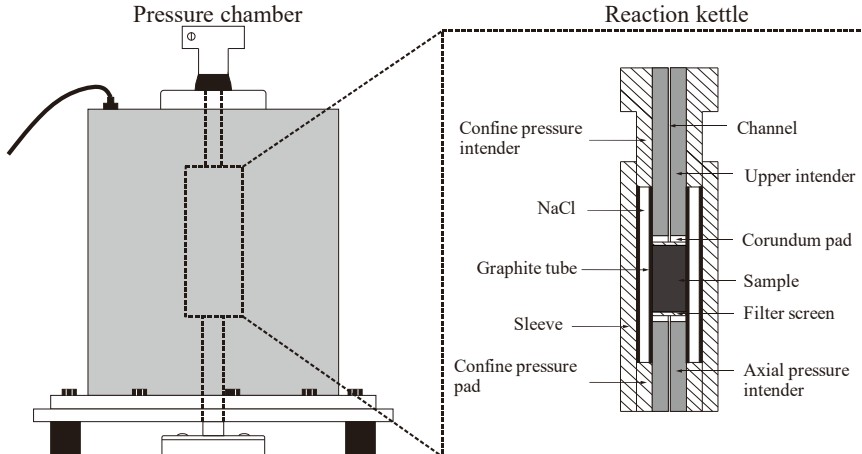

**Figure 2.** A schematic diagram of the deformation experimental system (modified from Liu et al. [14]).

**Table 2.** Experimental conditions for the coal samples.

| Sample Number | Confine Pressure (MPa) | Axial Pressure (MPa) | Temperature (°C) | Creep Experiment Time (h) | Deformation Types |
|---|---|---|---|---|---|
| 1 | 0.0 | 0.0 | 0.0 | 0.0 | Blank sample |
| 2 | 100.0 | 200.0 | 100.0 | 8.0 | Brittle- |
| 3 | 75.0 | 175.0 | 100.0 | 8.0 | Brittle- |
| 4 | 50.0 | 150.0 | 100.0 | 4.0 | Brittle- |
| 5 | 50.0 | 100.0 | 100.0 | 8.0 | Brittle- |
| 6 | 50.0 | 150.0 | 100.0 | 8.0 | Brittle- |
| 7 | 50.0 | 150.0 | 100.0 | 12.0 | Brittle- |
| 8 | 50.0 | 200.0 | 100.0 | 8.0 | Ductile- |
| 9 | 50.0 | 100.0 | 200.0 | 8.0 | Ductile- |
| 10 | 50.0 | 100.0 | 300.0 | 8.0 | Ductile- |

## 3. Results and Discussions

### 3.1. Deformation Behavior of Samples

Based on the structural–genetic classification scheme proposed by previous researchers, three deformation sequences could be clarified according to the micro deformation characteristics of the TDCs, viz. brittle, brittle–ductile and ductile deformation sequences [2,8,74]. The same site in the same sample was scanned by XRM after the deformation experiments for investigating the deformation behavior of the coal under the influence of stress and temperature. Among the experimentally deformed samples, samples 2–7 were classified as brittle deformed types, and samples 8–10 were classified as ductile deformed types in terms of deformation characteristics.

The samples in the brittle deformation sequence were mainly cut by multiple groups of fractures into cataclastic particles (Figure 3a–d). The width of fractures varied along the extension direction, which resulted in a necking or even thinning-out phenomenon. In sample 2, the width and density of the fractures were significantly smaller and lower than the other samples in the brittle deformed sequence, indicating that a higher confine pressure enhanced the strength of sample 2 and made it difficult for coal to be deformed by stress. Furthermore, fractures were found to be denser and more multidirectional in sample 5, subject to a lower confine pressure and longer deformation time, as compared to samples 2–3.

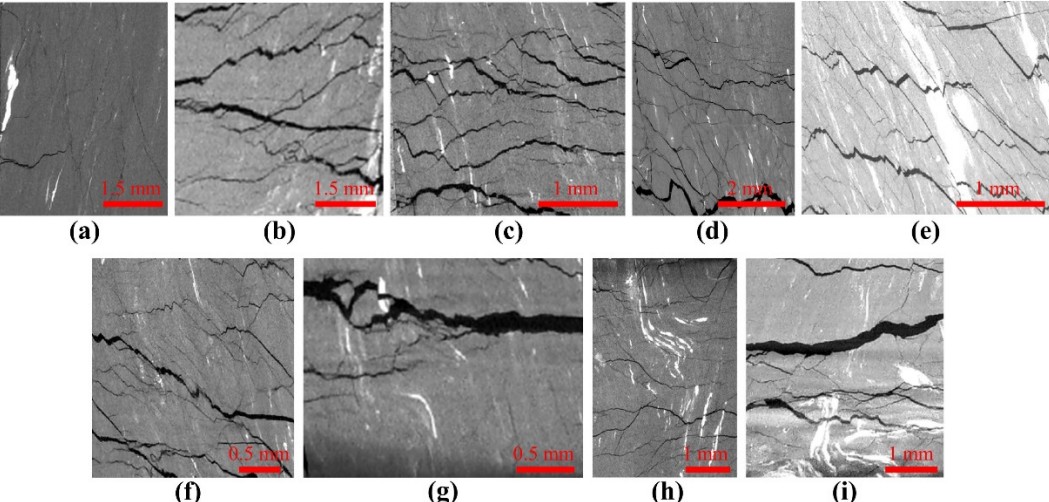

**Figure 3.** Deformation characteristics of the ductile deformed coals. (**a**) Sample 2, (**b**) Sample 3, (**c**) Sample 4, (**d**) Sample 5, (**e**) Sample 6, (**f**) Sample 7, (**g**) Sample 8, (**h**) Sample 9, (**i**) Sample 10.

For samples 6 and 7, a set of parallel fractures dominated the deformation structures (Figure 3e,f), which is consistent with the deformation characteristics of naturally developed TDCs [60,75]. The extension and width of the set of parallel fractures in sample 6 were

relatively stable. The coal matrix slides along fracture surfaces with displacements, which leads to the development of striae friction surfaces (Figure 3f). A series of accessory fractures next to the main fractures could be observed in sample 7. With the increase in the deformation time, the fracture density showed an increasing trend in sample 7 as compared with sample 6. Furthermore, the angles between the dominant fractures and the axial stress ($\sigma_1$) increased in sample 7 as well.

Small-scale superimposed folds developed in coals indicate shear ductile deformation [14,76,77]. In samples 8–10, diverse types of small folds were formed under the extrusion axial stress, which indicates the development of ductile deformation (Figure 3g,h). With the increase in the deformation intensity, the morphological characteristics of small folds become more complex. The small folds in sample 8 subject to a lower temperature were broad and symmetrical. In sample 9, a composite fold composed of several secondary folds could be observed. As for sample 10, the fold structures were more commonly developed, and a tight fold with a limb angle smaller than 30° could be found. Therefore, the mechanical properties of coals are significantly changed with the increase in temperature, which increases the plasticity of coals. In addition, it should be noted that, based on the various deformation characteristic of samples 6 and 8 subject to different differential stress and samples 4 and 9 subject to different temperatures, the transition conditions from brittle to ductile deformation can be classified as a differential stress of 100–150 MPa and a temperature of 100–200 °C.

### 3.2. Evolution of HBs

#### 3.2.1. Calculation

Characteristic sub-peaks of HBs could be derived from the FTIR spectrums in the range of 2400–3600 cm$^{-1}$ utilizing the Gaussian deconvolution method in Origin 7.5 software (Electronic Arts, Redwood City, CA, USA), which is precise enough for the strength and content calculation of HBs (Figure 4) [65]. Seven types of HBs could be identified in the selected coals, viz. SH–OH, COOH dimers, OH–N, cyclic OH, OH–ether, self-associated n-mers (n > 3) and OH–π (Figure 4 and Table 3) [44].

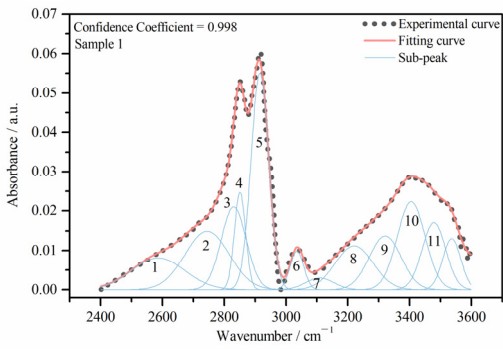

**Figure 4.** Curve fitting of the FTIR spectrum ranging from 2400 cm$^{-1}$ to 3600 cm$^{-1}$. (Assignment of sub-peaks (1−11) can be found in Table 3).

**Table 3.** Assignment of the FTIR spectrum in the range of 2400–3600 cm$^{-1}$ (According to Painter et al. [78], Miura et al. [65] and Li et al. [79]).

| Peak Number | Types of HBs | Band Positions (cm$^{-1}$) |
| :---: | :---: | :---: |
| 1 | SH–OH | ~2508.0 |
| 2 | COOH dimers | ~2739.0 |
| 3–5 | CH of aliphatic structures | — |
| 6 | OH–N | ~3006.0 |
| 7 | CH of aromatic structures | ~3050.0 |

**Table 3.** *Cont.*

| Peak Number | Types of HBs | Band Positions (cm$^{-1}$) |
|:---:|:---:|:---:|
| 8 | Cyclic OH | ~3200.0 |
| 9 | OH–ether | ~3300.0 |
| 10 | Self-associated n-mers (n > 3) | ~3406.0 |
| 11 | OH–π | ~3506.0 |

The amount of HBs ($N_{ab}$) could be calculated based on the Lambert–Beer law as following:

$$N_{ab} = \frac{A_{ab}}{I_{ab}} \tag{1}$$

where $a$ and $b$ refer to samples with different experimental conditions (from sample 1 to sample 10) and diverse types of HBs (seven types of HBs, from SH–OH to OH–π) respectively, and $A_{ab}$ and $I_{ab}$ refer to the integral intensity and absorptivity of the $b$ type of HB in sample $a$ with specific experimental conditions, respectively. According to Miura et al. [65], Equation (1) can be transformed into Equation (3) through the given Equation (2) of $I_{ab}$:

$$I_{ab} = I_0(1 + 0.0147 \times \Delta_{vOH,ab}) \tag{2}$$

$$N_{ab} = \frac{A_{ab}}{I_0(1 + 0.0147 \times \Delta_{vOH,ab})} \tag{3}$$

where $I_0$ refers to the absorption coefficient of free OH groups, and $\Delta_{vOH,ab}$ refers to the wavenumber shift of the $b$ type of HB relative to that of the free OH groups (Li et al. [44]). Therefore, the total amount of all types of HBs can be calculated utilizing Equation (4).

$$N_{total} = \sum_{ab} \frac{A_{ab}}{I_0(1 + 0.0147 \times \Delta_{vOH,ab})} \tag{4}$$

Furthermore, the strength of HBs can be estimated by the enthalpy ($\Delta H < 0$) of HB formation reactions. According to Miura et al. [65], the strength of the $b$ type of HB in sample $a$ with specific experimental conditions, $-\Delta H_{ab}$, can be calculated by Equation (5):

$$-\Delta H_{ab} = 0.067 \times \Delta_{vOH,ab} + 2.64 \tag{5}$$

Correspondingly, the average strength ($-\Delta H_{av}$) and total strength ($-\Delta H_{total}$) of HBs can be calculated as the following:

$$-\Delta H_{total} = \sum_{ab} \frac{A_{ab}}{I_0(1 + 0.0147 \times \Delta_{vOH,ab})}(0.067 \times \Delta_{vOH,ab} + 2.64) \tag{6}$$

$$-\Delta H_{av} = \frac{\sum_{ab} \frac{A_{ab}}{I_0(1+0.0147 \times \Delta_{vOH,ab})}(0.067 \times \Delta_{vOH,ab} + 2.64)}{\sum_{ab} \frac{A_{ab}}{I_0(1+0.0147 \times \Delta_{vOH,ab})}} \tag{7}$$

3.2.2. Content of HBs

The content of each type of HB and the total content of HBs in the TDCs were calculated utilizing Equations (3) and (4), respectively. As shown in Figure 5a–c, the content of the first six types of HBs, viz. SH–OH, COOH dimmers, OH–N, cyclic OH, OH–ether and self-associated n-mers (n > 3), all gradually decrease with the increase in the coal deformation intensity. However, HBs as a kind of weak bond as compared with covalent bonds could not only be easily broken but be made even at ambient temperature [80]. The content of OH–π with the lowest strength increases with the increase in coal deformation intensity, which is mainly ascribed to the faster formation rate of OH–π than the breaking rate during the deformation process (Figure 5d) [79]. In addition, the total content of HBs first decreases

from sample 1 to sample 6 and then rises from sample 6 to sample 10. In the decreasing stage of the total content, the formation rate of OH–π is much lower than the breakage rate of HBs in brittle deformed samples 2–5. In the increasing stage of the total content, the formation rate of OH–π increases because of the promotion of the stronger brittle deformation of samples 6 and 7, and the ductile deformation and higher temperature of samples 8–10. Even so, the total HB content of samples 6–10 is still lower than that of samples 2–5 in the decreasing stage, indicating that only a small part of the disassociated HBs was transformed into OH–π [81].

More specifically, the content of the first six types of HBs in samples 9 and 10 with a temperature of 200 and 300 °C was much lower in comparison with sample 5 with a temperature of 100 °C. HBs in coal without being subjected to stress treatment could be broken at temperatures higher than 150 °C [82]. Therefore, the disassociation rate of HBs in samples 9 and 10 was significantly accelerated by the higher temperature. Equally, the higher differential stress also led to the disassociation of more HBs, as is evident by comparing samples 5, 6 and 8. Besides, various contents of HBs in samples 4 and 6 indicate that more mechanical energy is generated with the increase in the creep experiment time, resulting in the breakage of more HBs. In contrast, the HB content in samples 2 and 3 is higher than that in sample 6, subject to a lower confine pressure, indicating that the disassociation process of HBs is inhibited by confine pressure.

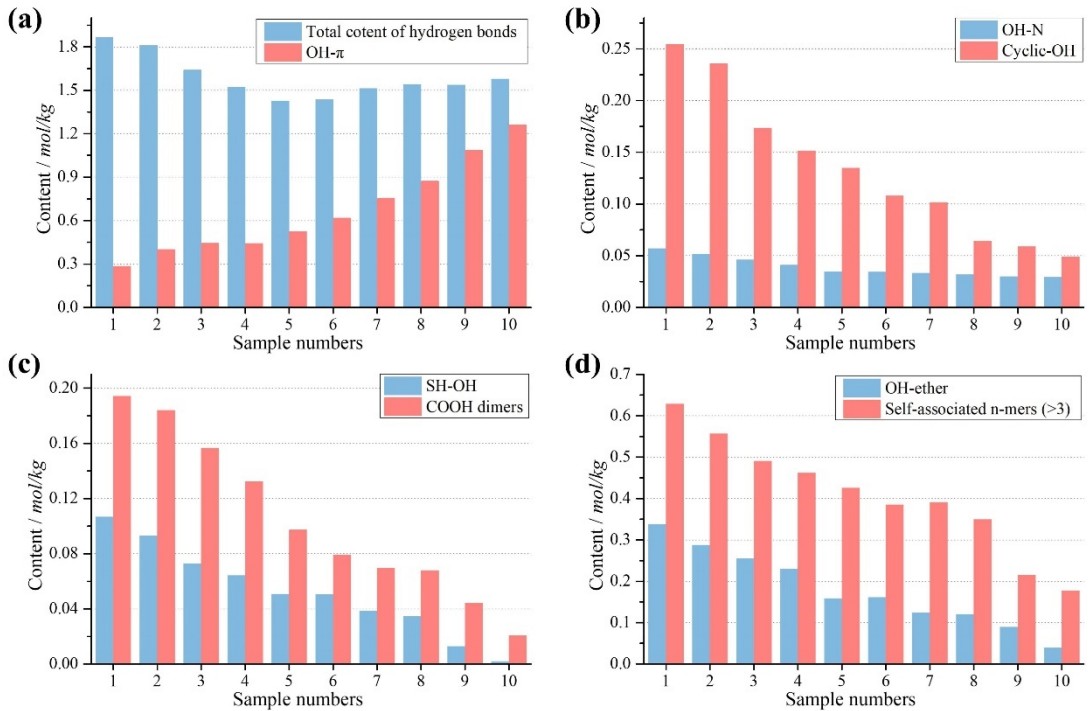

**Figure 5.** Content of HBs. (**a**) Content of SH–OH and COOH dimmers, (**b**) content of OH–N and cyclic OH, (**c**) content of OH–ether and self-associated OH, (**d**) content of OH–π and total content of HBs.

### 3.2.3. Strength of HBs
Distribution Patterns of HBs

The enthalpy for the formation reactions of different types of HBs was calculated using Equation (5). As shown in Figure 6, the average content and formation enthalpy of samples 2–7 were calculated to characterize the bond strength distribution pattern of the brittle deformed coals. Similarly, the average values of samples 8–10 were used to represent the ductile deformed coals. The strength of all seven types of HBs is in the range of 0–80 kJ/mol, which is much lower than that of covalent bonds. The content of HBs in the higher enthalpy region (>15 kJ/mol) gradually decreases from the primary coal

to the ductile deformed coals (Figure 6). However, the content of OH–π located in the lower enthalpy region (<15 kJ/mol) significantly increases with the increase of deformation intensity. It can be indicated by the complementary variation of HBs in the lower and higher enthalpy regions that there is a rearrangement among HBs under the influence of deformation and heating. After the cleavage, HBs in the higher enthalpy region are inclined to jump to the binding sites of aromatic structures, which is mainly caused by the rearrangement of more mobile aromatic structures and a faster formation rate [79,81].

For the same HB, the bond strength primarily depends on the length and angle of the HBs [83]. Overall, the strength of HBs retained or formed in the deformation process all show an increasing trend from the primary coal to the ductile deformed coals (especially the ductile deformed ones), indicating that the length of HBs is significantly compressed to shorter ones by differential stress.

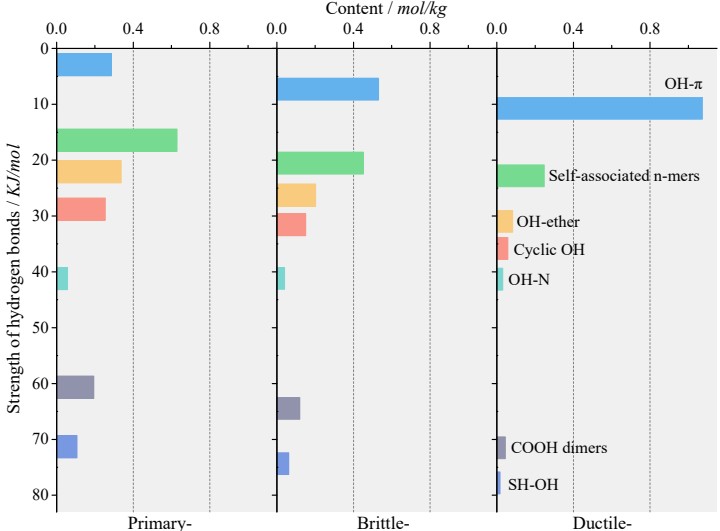

**Figure 6.** Distribution of HB strength in primary coal and the other two deformation sequences.

Total and Average Enthalpy of HBs

Equations (6) and (7) were used to calculate the total and average enthalpy for the formation reactions of HBs, respectively. Although the strength of each type of HB is enhanced in the deformed samples, the calculation results of these two parameters showed that both the total and average enthalpy gradually decrease with the increase in the coal deformation intensity, which is mainly attributed to the disruption of HBs (except for OH–π) and the transformation between OH–π and the other HBs under the coupling function of stress and temperature (Figure 7).

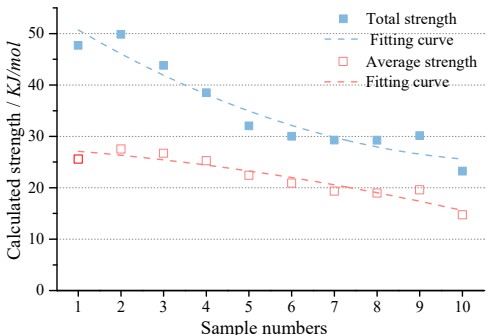

**Figure 7.** Total and average strength of HBs.

More concretely, the total and average enthalpy slightly decrease from sample 5 to samples 6 and 8 with the increase in the differential stress, indicating that differential stress advances the disruption reaction of HBs. Conversely, a comparison among samples 2, 3 and 5 indicates that the confine pressure significantly hinders the disruption reactions of HBs. Furthermore, the total and average enthalpy both sharply decline in sample 10 with a temperature of 300 °C as compared with samples 5 and 9, which is mainly ascribed to the HBs' heating disassociation and transformation to OH–π with lower enthalpy.

### 3.3. Functional Groups and $\pi-\pi$ Bonds

### 3.3.1. Curve Fitting of XPS Spectrums

The XPS spectra of coal samples contain important information about coal molecular structures [84]. In this study, spectra were deconvoluted by Origin 7.5 software (Developed by OriginLab corporation, Massachusetts, USA) (Figure 8) in the range of 280–292 cm$^{-1}$ (C1s), providing useful information about π–π bonds and oxygen functional groups [68,69]. The assignment of sub-peaks is presented in Table 4 in detail.

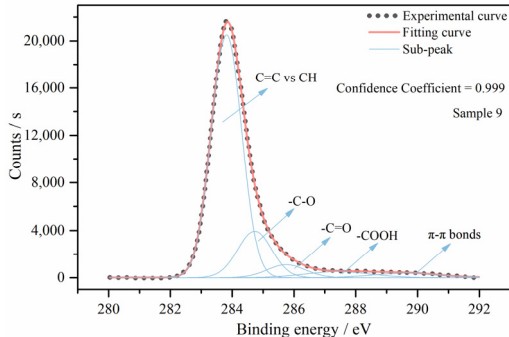

**Figure 8.** Curve fitting of the XPS spectrum ranging from 280 eV to 292 eV.

**Table 4.** Assignment of C(1s) peaks (according to Perry and Grint [68] and Shi et al. [69]).

| Binding Energy (eV) | Assignment |
| --- | --- |
| ~285.0 | Aromatic or aliphatic carbons |
| ~286.3 | Ether and hydroxyl |
| ~286.6 | Carbonyl |
| ~289.2 | Carboxyl |
| ~290.5 and ~291.5 | π–π bonds between aromatic species |

### 3.3.2. Variation of Functional Groups

In the selected coal samples, the -C-O type of groups including ether and phenol dominate the functional groups (Figure 9a), while the other two types of functional groups, -C=O and -COOH, have similar content grades. Three types of oxygen functional groups, as important hydrogen donors and acceptors of HBs, generally decrease with increasing deformation intensity, which can directly result in the loss of HBs. However, it was found that the variation of oxygen functional groups was not completely consistent with the variation of HBs related to those groups, indicating that the disruption of HBs is due partly to the disassociation of oxygen functional groups. Probably, the stress-induced geometry alteration of HBs can directly break them without attacking oxygen functional groups.

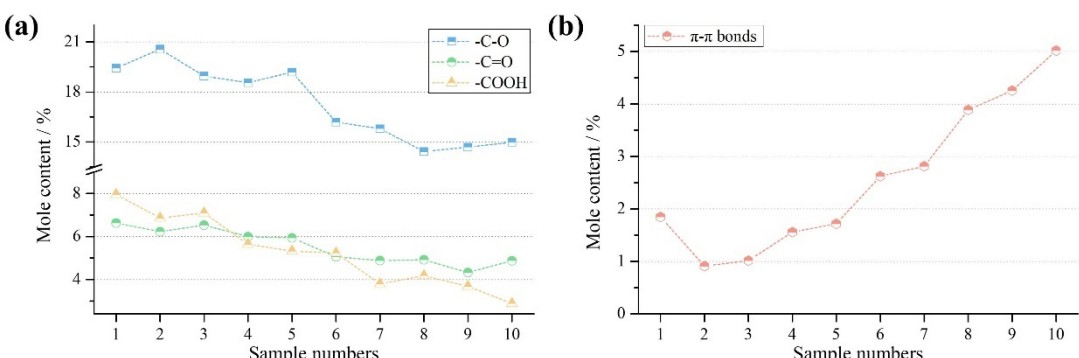

**Figure 9.** Fitting results of the XPS spectra, (**a**) mole content distribution of the three types of functional groups, (**b**) distribution of π–π bonds.

### 3.3.3. Differentiation of π–π Bonds

As shown in Figure 9b, the content of π–π bonds in the brittle deformed ones is all lower than that of primary coal; nevertheless, the content of the π–π bonds in samples 6, 7 and the ductile deformed coals are higher than the primary coal. In the view of previous researchers, the number of π–π bonds in coal is decided by the amount and configuration of aromatic structures. More aromatic structures with a better configuration (face-to-face configuration) commonly lead to the formation of more π–π bonds [60]. On the one hand, coal ductile deformation is generally considered to promote the aromatization and aromatic polycondensation reactions [23,24]; on the other hand, ductile deformation is better in favor of the motion of aromatic structures [60,85]. Therefore, compared with sample 1, the content of π–π bonds significantly increases in samples 8–10. However, it should be noted that π–π interactions between aromatic species less than 41.81 kJ/mol are in a dynamic variation process, and they can be disassociated in the structural relaxation process or formed within the face-to-face stacked aromatic structures [46,60]. The disassociation rate is higher than the formation rate of π–π bonds in brittle deformed samples 6 and 7, which is ascribed to the stronger brittle deformation compared with samples 2–5.

Considering the specific deformation conditions, samples 2 and 3, subject to a higher confine pressure than sample 4, have a lower content of π–π bonds, indicating that confine pressure restrains the chemical alteration process [23]. However, the content of π–π bonds increases sharply from samples 5 and 6 to sample 8, with a differential stress up to 150 MPa, which shows that aromatic structures are condensed or rearranged under the influence of differential stress. As for deformation temperatures, the content of π–π bonds in samples 9 and 10, which is twice as much as that in sample 5, illustrates that the coalification process of the samples is significantly advanced with the increase in temperature.

### 3.4. Order Degree of Coal Molecular Structures

Two distinct humps could be observed in the first order region (1000 cm$^{-1}$ to 1800 cm$^{-1}$) in the Raman spectra (Figure 10). The first hump at around 1350 cm$^{-1}$ represents the disorder structures, mainly including amorphous carbons, secondary structural defects and even hetero atoms (the so-called disorder-induced band or D band) [86]. The second broad hump at around 1600 cm$^{-1}$ represents the graphene structures in coal (the so-called G band). The ratio of $I_D/(I_D + I_G)$ is an important parameter to characterize the order degree of coal molecular structures, which are determined by the proportions of aromatic layers with face-to-face configuration and disordered amorphous carbons [87,88]. The intensities of the G and D bands were obtained using the Gaussian fitting method in Origin 7.5 software.

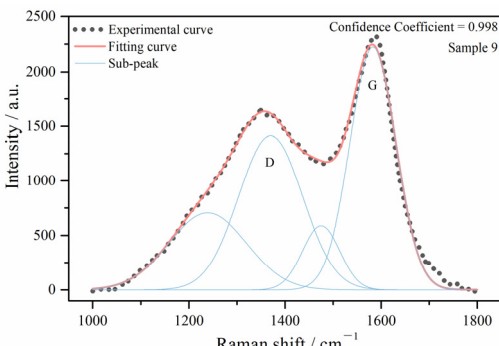

**Figure 10.** Curve fitting of Raman spectrum ranging from 1000 cm$^{-1}$ to 1800 cm$^{-1}$. (D refers to the D band, and G refers to the G band).

The ratios of $I_D/(I_D + I_G)$ first increase from samples 1 to 3 and then decrease from samples 4 to 10 (that of samples 9 and 10 are even lower than that of the primary coal), which indicates that the order degree of the coal molecular structures shows a decreasing trend in samples 1, 2 and 3, while the molecular structures of samples 4 to 10 gradually begin to rearrange to more ordered ones (Figure 11). However, the rearrangement of the coal molecular structures to an ordered configuration is a reversible process. Thus, for samples 2–8, the relaxation rate of the coal's molecular structures resulting from the breakage of non-covalent bonds is higher than the rearrangement rate, which leads to the $I_D/(I_D + I_G)$ of samples 2–8 being lower than that of the primary coal. Meanwhile, the rearrangement rate in samples 9 and 10 is much higher than the relaxation rate, resulting in a significant increase in the proportion of ordered coal molecular structures with face-to-face configurations.

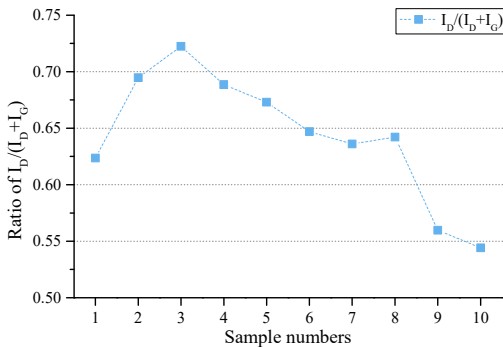

**Figure 11.** Distribution of $I_D/(I_D + I_G)$ ratios.

### 3.5. Evolutionary Patterns of Coal Network Structures

Considering the variation of non-covalent bonds and the order degree of coal molecular structures, it is not difficult to conclude that the stress-induced and thermal evolutionary patterns of coal molecular structures can be classified into two main categories, viz. relaxation and rearrangement (Figure 12). During the relaxation process, the aromatic structures are relaxed through the breakage of non-covalent bonds, which is demonstrated by the images of aromatic lattice fringes in sample 6 and the ratios of $I_D/(I_D + I_G)$) calculated from the Raman detection results (Figure 12, stage b). Six types of HBs including SH–OH, COOH dimmers, OH–N, cyclic OH, OH–ether, self-associated n-mers (n > 3) and even $\pi$–$\pi$ bonds are disassociated by mechanical and heating treatment. Part of the disassociated HBs is transformed into OH–$\pi$ with lower strength. Thus, the total and average strength of HBs are weakened by deformation. The mobility of coal molecular structures significantly increases with the alteration of non-covalent bonds, which enables the mobile phases to move or be rearranged, provided that the moving barrier is overcome.

However, as for the rearrangement process, the directions of the lattice fringes tend to be consistent with each other, which illustrates that a more ordered configuration of coal molecules is formed in the ductile deformed sequence (Figure 12, stage c). The strength of most HBs is enhanced, and the number of π–π bonds increases with the increase in the deformation intensity, which leads to the formation of a more ordered and tighter configuration. The structural variations between samples 9 and 10 show that a higher temperature significantly accelerates the breakage of non-covalent bonds and the rearrangement process. The two evolutionary patterns are not independent during the deformation process, that is, relaxation and rearrangement occur simultaneously under the influence of stress and heating. However, in brittle deformed coals, the relaxation process predominates the deformation process, resulting in less-ordered molecular configurations. Conversely, the evolution of ductile deformed coals is predominated by the rearrangement process, indicating that a tighter and more ordered configuration is generated.

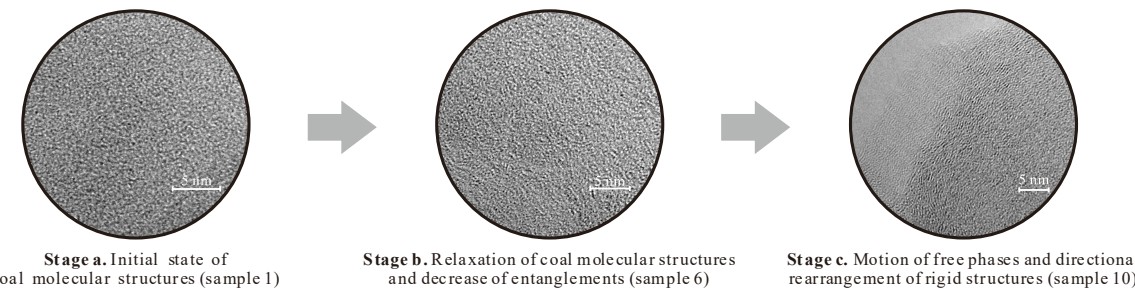

**Stage a.** Initial state of coal molecular structures (sample 1)  **Stage b.** Relaxation of coal molecular structures and decrease of entanglements (sample 6)  **Stage c.** Motion of free phases and directional rearrangement of rigid structures (sample 10)

**Figure 12.** Evolution path of coal molecular structures during deformation experiments.

### 3.6. Understanding of Coal Deformation Mechanisms from the Nano-Scale Perspective

3.6.1. Brittleness and Plasticity of Coals

Coal macromolecular structures are stabilized through the connection of covalent bonds, non-covalent bonds and the entanglements of networks [42]. The mechanical properties of coal are intimately correlated with the coal's chemical structures [89]. Investigations on the thermal plasticizing of coal indicate that the mechanical properties of coal are closely correlated with the existence of non-covalent bonds [54,58,90,91]. With increasing temperature, the entanglements of molecules and the cleavage of non-covalent bonds decrease and increase respectively, which in turn promotes the mobility of molecules and increases the rheological properties of coal [58]. Therefore, the plasticity of coals significantly increases with the disassociation of HBs and π–π bonds, though the viscoelastic properties decrease [92]. Conversely, the brittleness of coal increases with the increasing number of non-covalent bonds [54].

During the high temperature and pressure experiments, a bunch of non-covalent bonds was disrupted in stage b under the coupling effects of stress and temperature, which directly relaxed the network structures and produced a more mobile phase [56]. In the brittle deformed sequence, structural fractures and cleats spread along the belts in those samples with a weaker bond force (viz. non-covalent bonds), which finally leads to the typical cataclastic feature of brittle deformed coals [93]. Besides, the disruption of HBs in brittle deformed coals could not result in the transformation into ductile deformation, which is mainly attributed to two reasons. That is, the thermal and mechanical energy is not enough for molecules to overcome the motion barrier energy, and there is a lack of enough time for the motion of liberated small molecules (sample 4 with deformation time of 4 h, sample 7 with deformation time of 12 h and sample 6 with deformation time of 8 h).

Conversely, micro folds of maceral bands emerged in samples 8, 9 and 10, subject to higher temperatures and differential stress, indicating that the plasticity of these three samples was significantly enhanced. More mechanical and thermal energy acts on coal molecular structures with the increase in differential stress and temperatures, which significantly advances the cleavage and transformation process of non-covalent bonds. With the

breakage of non-covalent bonds, the number of liberated molecules in ductile deformed coals notably increases as compared with brittle deformed coals. Therefore, the mobility of coal molecules is correspondingly raised, that is, the inner chemical structures in the coal are relaxed and allowed to move locally. Oriented moving and rearrangement of coal molecules directly lead to the transformation into OH–π and the formation of more π–π bonds. Strain energy transformed from ductile deformation is released through the inner motion of mobile molecules generated by the cleavage of non-covalent bonds. The macro-scopically ductile deformation of coals finally occurs through a series of inner structural adjustments of the coal molecules.

3.6.2. Transition Conditions from Brittle to Ductile Deformation

Pioneering results of coal deformation experiments have shown that there are key conditions for the transition from brittle to ductile deformation. As for coals with different ranks, the deformation transition temperature between brittle and ductile is different as well. The transition temperature for higher-rank coals ($R_{o, max}$ > 1.70%) is considered to be in the range of 200–300 °C [73]. The results of the current study show that the transition temperature for bituminous coal with a $C_{daf}$ of 84.35% ($R_{o, max}$ = 0.88%) is in the range of 100–200 °C. In addition, our previous study indicated that 100 °C was sufficient for long flame coal ($R_{o, max}$ = 0.67%) to transit from brittle to ductile deformation [14].

The coalification process is characterized by the disassociation of functional groups and the condensation of aromatic structures, which in turn results in the decrease in the number of HBs and an increase in the number of π–π bonds [94]. The configuration of aromatic structures in higher-rank coals is tighter and more ordered than those in lower-rank coals. Besides, the strength of π–π bonds is higher than most HBs. The upshot is that, with the increase in coal rank, the coal's molecular structures become more stable and tighter, and the coal's mechanical properties are strengthened [89]. Therefore, the dissociation energy of non-covalent bonds and the energy barrier of molecular motion both increase with the increase in coal rank. Accordingly, the transition temperature of ductile deformation is increased from low- to high-rank coal.

## 4. Conclusions

To investigate the deformation mechanism and role of non-covalent bonds in the deformation process, high temperature and pressure deformation experiments were performed on a TRTP 2000 experimental system, and the characteristics of samples before and after the deformation experiments were detected by FTIR, XPS, Raman spectroscopy and HRTEM.

The simulative experiments illustrate that higher differential stress, temperature and deformation time promote the deformation process from brittle deformation to ductile deformation. Conversely, confine pressure hinders the deformation process and enhances the mechanical strength of coal. The deformation transition differential stress and temperature for bituminous coal are in the ranges of 100–150 MPa and 100–200 °C, respectively.

The molecular evolution of deformed coals can be summarized into two main patterns from the view of the variation of coal's molecular structures, viz. relaxation and rearrangement stages. In the relaxation stage, the HBs and π–π bonds are more inclined to be disrupted with the increase of coal deformation, which promotes the motion of molecular structures. In the rearrangement stage, tighter and more ordered configurations are formed, accompanied by the formation of π–π bonds. Non-covalent bonds, both HBs and π–π bonds, as important crosslinks of coal molecular networks, determine the mechanical properties of coals. The disassociation of enough non-covalent bonds in the deformed coals formed under the transition deformation conditions leads to a decrease in the coal's mechanical strength and promotes the ductile deformation process.

Overall, the variation of the non-covalent bonds in experimental coals is in good agreement with those of the natural TDCs. In the current study, only the brittle and ductile deformed types of coals are formed in the simulative experiments, while, the brittle–ductile deformed type of coal, which is the key to obtain the specific transition conditions for coal

deformation, cannot be simulated under current conditions. Therefore, the simulation of brittle–ductile deformed type of coals will be the focus of our work in the future.

**Author Contributions:** H.L. prepared the manuscript and analyzed the data; C.H. undertook the experiments. All authors have read and agreed to the published version of the manuscript.

**Funding:** The research is sponsored by the National Natural Science Foundation of China (Nos. 41672147, 41430317, 42102221).

**Institutional Review Board Statement:** Not applicable.

**Informed Consent Statement:** Not applicable.

**Data Availability Statement:** The datasets used and/or analyzed during the current study are available from the corresponding author on reasonable request.

**Conflicts of Interest:** The authors declare no competing financial interest.

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
