# Peer review of "The Role of Non-Covalent Bonds in the Deformation Process of Coal: An Experimental Study on Bituminous Coal"

_processes, doi:10.3390/pr10091875_

Round 1

Reviewer 1 Report

The authors have presented an excellent paper that can be considered for publication in the journal. The proposed experiments, their implementation and the discussion of the results are coherent and present a good contribution for those interested in the area. For this reason, only a few minor comments are indicated which need to be addressed before final acceptance.

The introduction is very appropriate and provides a sufficient number of references. The only comment would be to improve the objective of the article a little, it is somewhat unclear and I think it could be better formulated.

Enlarge the size of Figure 2

In Table 2 and others, put the units in brackets. Example: Confine pressure (MPa), be careful with Pascals. This is extrapolable to the graphs.

The results are consistent and well discussed. Would it be possible to include the error bars in the graphs?

Use the same number of significant figures in all tables.

Conclusions should be expanded and contain possible limitations of the study conducted. In addition, the auotres should include all the sections of the journal Template.

In any case, this is an excellent article that deserves to be considered for publication.

Author Response

Dear editor Isabella Toth and reviewers,

Thank you for your comments and consideration concerning on our manuscript entitled "The role of non-covalent bonds in coal deformation process: An experimental study on bituminous coal" (processes-1903880). We have read comments very carefully and made corrections, which we hope to meet with the approval. Revised portions are marked red in the revised manuscript. If you have any other requirements, don't hesitate to contact us. We will be pleased to undertake the work required. These main corrections in the manuscript and the responses to the comments are as following,

Reviewer 1

The authors have presented an excellent paper that can be considered for publication in the journal. The proposed experiments, their implementation and the discussion of the results are coherent and present a good contribution for those interested in the area. For this reason, only a few minor comments are indicated which need to be addressed before final acceptance.

Issue 1: The introduction is very appropriate and provides a sufficient number of references. The only comment would be to improve the objective of the article a little, it is somewhat unclear and I think it could be better formulated.

Response

The main objective of this study is to reveal the micro deformation mechanism of coal by figuring out the role of non-covalent bonds in coal deformation process. Therefore, we rewrite the last paragraph of the introduction. The sentences “…In this study, to reveal micro coal deformation mechanism…” and “By connecting the mechanical properties of coal to the evolution of non-covalent bonds, the coal deformation mechanism can be better revealed at the nano-scale.” are added to clearly clarify the research objective of this article. Would you agree with us?

Issue 2: Enlarge the size of Figure 2

Response

The size of Figure 2 is now enlarged in the revised manuscript.

Issue 3: In Table 2 and others, put the units in brackets. Example: Confine pressure (MPa), be careful with Pascals. This is extrapolable to the graphs.

Response

All the units in the tables are now put in brackets, and the wrong units “Mpa” are corrected to “MPa” in the revised manuscript.

Issue 4: The results are consistent and well discussed. Would it be possible to include the error bars in the graphs?

Response

The error bars are very important in scientific researches as you mentioned. Unfortunately, since we have not carried out repeated simulative experiments and only one experiment has been carried out for each deformation condition. Besides, the experimental samples have been used up, it is impossible to supplement the repeated simulative experiments, so we cannot draw error bars in the graphs under current conditions. We will pay more attention to it in the follow-up study.

Issue 5: Use the same number of significant figures in all tables.

Response

The number of significant figures in all tables is unified in the revised manuscript.

Issue 6: Conclusions should be expanded and contain possible limitations of the study conducted.

Response

In the revised manuscript, the conclusions are expanded and the possible limitations of this study are also conducted in the last paragraph of the revised paper. The added last paragraph is “Overall, the variation of the non-covalent bonds in experimental coals is in good agreement with those of the natural TDCs. In the current study, only the brittle and ductile deformed types of coals are formed in the simulative experiments. While, the brittle-ductile deformed type of coal, which is the key to obtain the specific transition conditions for coal deformation, cannot be simulated under current conditions. Therefore, the simulation of brittle-ductile deformed type of coals will be the focus of our work in the future.”.

Issue 7: In addition, the authors should include all the sections of the journal Template.

Response

The missing “Institutional Review Board Statement”, “Informed Consent Statement”, and “Data Availability Statement” sections are all added in the revised manuscript according to the journal template.

In any case, this is an excellent article that deserves to be considered for publication.

Thank you very much for your comments and hard work, which have greatly helped us to improve this paper.

Kind regards.

Hewu Liu

2022.09.11

Reviewer 2 Report

The manuscript is very interesting and well-composed.

It contains well designed experimental system and produced results that should be useful for readers who work on this subject.

However, the manuscript needs to be checked before being accepted.

For example, the writing of the manuscript needs a serious check because of many grammatical mistakes.

Please see the attached file for my comments/corrections on the paper.

Overall, the manuscript can be accepted after minor corrections/revisions to be published in the journal of Processes.

Author Response

Dear editor Isabella Toth and reviewers,

Thank you for your comments and consideration concerning on our manuscript entitled "The role of non-covalent bonds in coal deformation process: An experimental study on bituminous coal" (processes-1903880). We have read comments very carefully and made corrections, which we hope to meet with the approval. Revised portions are marked red in the revised manuscript. If you have any other requirements, don't hesitate to contact us. We will be pleased to undertake the work required. These main corrections in the manuscript and the responses to the comments are as following,

Reviewer 2

The manuscript is very interesting and well-composed. It contains well designed experimental system and produced results that should be useful for readers who work on this subject. However, the manuscript needs to be checked before being accepted.

Issue 1: For example, the writing of the manuscript needs a serious check because of many grammatical mistakes.

Response

We have carefully checked and corrected the grammatical errors in the full text.

Issue 2: Please see the attached file for my comments/corrections on the paper.

Response

We have carefully read the attached files and revised the manuscript according to your comments/corrections, which is very helpful to modify our paper.

Thank you very much for your comments and hard work, which have greatly helped us to improve this paper.

Kind regards.

Hewu Liu

2022.09.11

Reviewer 3 Report

The authors investigated the role of noncovalent bonds in coal deformation process via an experimental study on bituminous coal. The study is interesting and suitable for publication in Processes.

1. The abstract is appropriate and well written.

2. I  dont think Figure 1 is necessary in the introduction.

3. Please, highlight the novelty of the study in the last paragraph of the introduction.

4. Please, provide a better quality of Figure 2.

5. A brief description of the coal deformation experiment should be provided.

6. The authors should provide a high resolution image for Figure 4.

7. The results need to be discussed in comparison with literature. 

8. Please, carefully proofread for any language error.

Author Response

Dear editor Isabella Toth and reviewers,

Thank you for your comments and consideration concerning on our manuscript entitled "The role of non-covalent bonds in coal deformation process: An experimental study on bituminous coal" (processes-1903880). We have read comments very carefully and made corrections, which we hope to meet with the approval. Revised portions are marked red in the revised manuscript. If you have any other requirements, don't hesitate to contact us. We will be pleased to undertake the work required. These main corrections in the manuscript and the responses to the comments are as following,

Reviewer 3

The authors investigated the role of noncovalent bonds in coal deformation process via an experimental study on bituminous coal. The study is interesting and suitable for publication in Processes.

Issue 1: The abstract is appropriate and well written.

Response

Thank you very much.

Issue 2: I don’t think Figure 1 is necessary in the introduction.

Response

Figure 1 is deleted in the revised manuscript.

Issue 3: Please, highlight the novelty of the study in the last paragraph of the introduction.

Response

We believe that the novelty of this study is mainly manifested in two aspects: Firstly, we take the less studied non-covalent bond as our research object; Secondly, the coal deformation mechanism is revealed based on the evolution of non-covalent bonds. Therefore, we add the sentences “Consequently, the variation of non-covalent bonds during the coal deformation pro-cess has rarely been studied by researchers, especially by using simulative experiments.” and “By connecting the mechanical properties of coal to the evolution of non-covalent bonds, the coal deformation mechanism can be better revealed at the nano-scale.” in the last paragraph of the introduction to highlight the novelty. Would you agree with us?

Issue 4: Please, provide a better quality of Figure 2.

Response

The size of Figure 2 is enlarged and its quality is better now in the revised manuscript.

Issue 5: A brief description of the coal deformation experiment should be provided.

Response

A brief description of the coal deformation experiment is added in section 2.2.1 (Line241-248) in the revised manuscript.

Issue 6: The authors should provide a high resolution image for Figure 4.

Response

We have done our best to increase the resolution of Figure 4, and the picture clarity has been improved to some extent. However, due to the accuracy of the experimental instrument, the resolutions of Figs. 3b and 3g are still a little vague.

Issue 7: The results need to be discussed in comparison with literature.

Response

The comparison between our experimental results and literature is very important to explain the results. While, similar researches about coal deformation experiments are very limited, especially refer to the non-covalent bonds. Thus, it is a little hard to make comparison with literature. Despite all this, there are still some references being cited and compared in section 4.

For example, in section 4.1, the deformation behavior of coals is mainly compared with that of natural deformed coals (the references [2], [8], [62], [76], [77], [78], [79]) and experimental deformed coals in our previous research [14]. Herein, the above mentioned references are cited and compared to our research results in section 4.1.

In sections 4.2 and 4.3, because the variation of non-covalent bonds in deformed coals has rarely been investigated by other researchers. Only our previous research (reference [62]) talked about the evolution of non-covalent bonds in natural deformed coals. Therefore, [62] is one of the most cited references in this section. Besides, some other researches about the non-covalent bonds in coal (not deformed coals) are also cited and compared in sections 4.2 and 4.3, such as the references [48], [81], [82], [83], [84] etc.

Section 4.5 is the summary of sections 4.1 - 4.4, therefore, we believe that the comparison with literature does not need in this section.

Section 4.6 mainly discusses the coal deformation mechanism according to the experimental results. In this section, abundant references are cited and compared with our experimental results, for example, references [68], [14], [91], [97] etc.

Therefore, we think that the results have been fully compared with the literature. Would you agree with us?

Issue 8: Please, carefully proofread for any language error.

Response

The full text has been carefully checked and corrected with native English. And the revised words or sentences are marked red in the revised manuscript.

Thank you very much for your comments and hard work, which have greatly helped us to improve this paper.

Kind regards.

Hewu Liu

2022.09.11
